# Prostate Cancer in Renal Transplant Recipients: Results from a Large Contemporary Cohort

**DOI:** 10.3390/cancers15010189

**Published:** 2022-12-28

**Authors:** Giancarlo Marra, Francesco Soria, Federica Peretti, Marco Oderda, Charles Dariane, Marc-Olivier Timsit, Julien Branchereau, Oussama Hedli, Benoit Mesnard, Derya Tilki, Jonathon Olsburgh, Meghana Kulkarni, Veeru Kasivisvanathan, Cedric Lebacle, Oscar Rodriguez-Faba, Alberto Breda, Timo Soeterik, Giorgio Gandaglia, Paola Todeschini, Luigi Biancone, Paolo Gontero

**Affiliations:** 1Department of Surgical Sciences, University of Turin and Città della Salute e della Scienza, 10126 Turin, Italy; 2Department of Urology, Institut Mutualiste Montsouris and Université Paris Descartes, 75014 Paris, France; 3Department of Urology, Hôpital Tenon, 75020 Paris, France; 4Department of Urology, Hôpital Européen Georges Pompidou, 75015 Paris, France; 5Institut de Transplantation Urologie Nèphrologie (ITUN), CHU Nantes, 44093 Nantes, France; 6Nuffield Department of Surgical Sciences, Oxford University, Oxford OX1 2JD, UK; 7Martini-Klinik Prostate Cancer Center, University Hospital Hamburg-Eppendorf, 20251 Hamburg, Germany; 8Department of Urology, University Hospital Hamburg-Eppendorf, 20251 Hamburg, Germany; 9Department of Urology, Guy’s Hospital, London SE1 9RT, UK; 10University College London, London WC1E 6BT, UK; 11Department of Urology, Kremlin-Bicêtre Hospital, 94270 Le Kremlin-Bicêtre, France; 12Department of Urology, Fundacio Puigvert, 08025 Barcelona, Spain; 13Department of Urology, Saint Antonius Hospital, 3543 AZ Utrecht, The Netherlands; 14Department of Urology, San Raffaele Hospital, 20132 Milan, Italy; 15Department of Nephrology, Sant’Orsola Malpighi Hospital, 40138 Bologna, Italy; 16Department of Nephrology, University of Turin and Città della Salute e della Scienza, 10126 Turin, Italy

**Keywords:** prostate cancer, renal transplant, treatment, robotic radical prostatectomy, immunosuppression

## Abstract

**Simple Summary:**

Currently, low-level and contrasting evidence exists to guide the management of prostate cancer diagnosed in renal transplant recipients. The authors aimed to assess whether PCa treatment and/or natural history differed when diagnosed in RTRs. Overall, they found that PCa did not seem aggressive and PCa outcomes were similar to available evidence in non-RTRs, although RTRs had a non-negligible risk of non-PCa-related death. The authors concluded that indiscriminate aggressive upfront PCa management in RTRs should be avoided.

**Abstract:**

Objectives: The aim of this study was to assess the natural history of prostate cancer (PCa) in renal transplant recipients (RTRs) and to clarify the controversy over whether RTRs have a higher risk of PCa and poorer outcomes than non-RTRs, due to factors such as immunosuppression. Patients and Methods: We performed a retrospective multicenter study of RTRs diagnosed with cM0 PCa between 2001 and 2019. Primary outcomes were overall (OS) and cancer-specific survival (CSS). Secondary outcomes included biochemical recurrence and/or progression after active surveillance (AS) and evaluation of variables possibly influencing PCa aggressiveness and outcomes. Management modalities included surgery, radiation, cryotherapy, HIFU, AS, and watchful waiting. Results: We included 166 men from nine institutions. Median age and eGFR at diagnosis were 67 (IQR 60–73) and 45.9 mL/min (IQR 31.5–63.4). ASA score was >2 in 58.4% of cases. Median time from transplant to PCa diagnosis was 117 months (IQR 48–191.5), and median PSA at diagnosis was 6.5 ng/mL (IQR 5.02–10). The biopsy Gleason score was ≥8 in 12.8%; 11.6% and 6.1% patients had suspicion of ≥cT3 > cT2 and cN+ disease. The most frequent management method was radical prostatectomy (65.6%), followed by radiation therapy (16.9%) and AS (10.2%). At a median follow-up of 60.5 months (IQR 31–106) 22.9% of men (n = 38) died, with only n = 4 (2.4%) deaths due to PCa. Local and systemic progression rates were 4.2% and 3.0%. On univariable analysis, no major influence of immunosuppression type was noted, with the exception of a protective effect of antiproliferative agents (HR 0.39, 95% CI 0.16–0.97, p = 0.04) associated with a decreased risk of biochemical recurrence (BCR) or progression after AS. Conclusion: PCa diagnosed in RTRs is mainly of low to intermediate risk and organ-confined at diagnosis, with good cancer control and low PCa death at intermediate follow-up. RTRs have a non-negligible risk of death from causes other than PCa. Aggressive upfront management of the majority of RTRs with PCa may, therefore, be avoided.

## 1. Introduction

Prostate cancer (PCa) remains the most frequent non-skin solid neoplasm in men. This remains true in those who underwent a kidney transplant (RTR) [1,2].

With the constant increase in the number of transplants performed and the notable increase in the life expectancy of these patients, now almost reaching 20 years for recipients in their 50s [3,4,5], a rise in PCa cases diagnosed in RTRs is likely in the near future. 

In this context of improved healthcare around RTRs, knowledge of the natural history of PCa remains overall scarce. Evidence is low and mainly based on retrospective, relatively old series. 

As a consequence, several questions remain open, notably the preferred treatment option if any, whether these patients may have increased complication rates, and whether PCa management should be tailored differently in RTRs. Specifically, we do not clearly know whether RTRs have worse PCa features and oncological outcomes compared to the general population. Nonetheless, there is a tendency to actively treat even low-risk disease. Such practice seems due to a fear, not supported by any evidence, that an immunosuppression regimen to prevent graft rejection may decrease PCa immune control and, thus, favor disease progression [6,7].

This tendency for overtreatment also sharply contrasts with the lower life expectancy of RTRs and with some immunosuppression regimens used as chemotherapeutic agents for several urological and non-urological malignancies [5,6].

Two systematic reviews recently suggested no major differences in terms of aggressiveness when PCa is diagnosed in RTRs [5,8]. Population-based case–control studies also found PCa to not differ between RTRs and non-RTRs after propensity score matching, with the exception of lower overall survival in RTRs [9,10].

Nonetheless, current evidence from non-population-based cohorts relies on small samples, with the largest multi-institutional series including less than 100 patients [11].

Hence, we aimed to describe the natural history of PCa in a large multicenter contemporary cohort of men diagnosed with PCa after kidney transplant and to investigate possible factors influencing oncological control.

## 2. Materials and Methods

### 2.1. Data Collection

We retrospectively collected data of patients being diagnosed with histologically documented PCa after kidney transplant at nine European tertiary referral centers between 2001 and 2019. All cases were reviewed and updated from physician-maintained databases at each involved institution. At the time of kidney transplant, all patients underwent routine PSA and digital rectal examination (DRE), as well as prostate biopsies to rule out the presence of PCa when indicated.

At the time of PCa diagnosis, all patients underwent staging according to EAU guidelines (axial abdominal imaging—mpMRI and/or CT scan and bone scan). Fifteen patients also had staging PET scans negative for systemic extension (choline n = 12; PSMA n = 3). Two physicians independently performed the data quality review (G.M. and F.P.). Centers were re-contacted for data revision in cases of uncertainty or missing information. Metastatic patients were excluded (n = 5). 

### 2.2. Outcomes

The primary outcome was to describe overall survival (OS) and cancer-specific survival (CSS) of cM0 PCa in kidney transplant recipients. Secondary outcomes were to assess (i) BCR/progression rates, and (ii) kidney transplant, patient, and PCa factors possibly influencing OS, BCR/progression, and PCa histological aggressiveness.

### 2.3. Categorization of the Variables

Preoperative comorbidity status was recorded using the ASA score. Family history of PCa was defined as ≥2 first- or second-degree relatives with PCa on the same side of the pedigree.

Biochemical recurrence (BCR) was defined as an undetectable PSA subsequently reaching >0.2 ng/mL and rising after radical prostatectomy, with a rise by 2 ng/mL or more above the nadir PSA (ASTRO-Phoenix Criteria) after radiotherapy and, in the absence of validated criteria, after cryotherapy and HIFU. Local and/or systemic progression was defined as the presence of clinical recurrence assessed with any imaging modality. For the purpose of the analysis, we defined the event BCR/progression as BCR (after radiotherapy or radical prostatectomy or cryotherapy or HIFU) and/or progression from active surveillance (AS) to active treatment; watchful waiting and androgen deprivation therapy patients were not included in this analysis.

### 2.4. Statistical Analysis

Categorical and continuous variables were reported as absolute numbers/proportions and medians with interquartile ranges (IQRs). Chi-square and Kruskal–Wallis tests were performed for categorical and continuous variables to compare the populations, respectively. Univariable logistic regression analyses were performed to evaluate the impact of patient, kidney transplant, baseline, and PCa features on the probability of harboring a Gleason score ≥8 PCa at diagnosis. Univariable Cox regression analysis was performed to evaluate possible predictors of BCR or progression after AS and OS. Statistical analyses were performed using STATA 13 (Stata Corp., College Station, TX, USA). All tests were two-sided, and *p* < 0.05 was considered as statistically significant.

## 3. Results

### 3.1. Baseline Features

We included 166 men. Table 1 displays the baseline patients and kidney transplant features. Median age and BMI were 67 (IQR 60–73) years and 26 (IQR 24–29), respectively; n = 34 (24.1%) patients had a history of malignancies other than PCa. At PCa diagnosis, the median renal function of transplanted patients was 45.9 mL/min (IQR 31.5–63.4). Renal failure was chronic (98.6%) in the majority of cases, with chronic glomerulonephritis being the commonest prevailing cause (34.7%). Transplants were most frequently single (84.9%) and from a cadaver donor (82.2%). Calcineurin inhibitors were the most frequently used immunosuppression drug (70.5%).

### 3.2. PCa Features and Treatment

Table 2 shows the PCa features at diagnosis. The median time from transplant to PCa diagnosis and median PSA were 117 months (IQR 48–191.5) and 6.5 ng/mL (IQR 5.02–10), respectively. The biopsy Gleason score was ≥8 in 12.8%. Preoperative mpMRI was performed in 65.2%. Ten patients (6.1%) had a suspicion of cN+ disease at first diagnostic imaging.

The majority (n = 109, 65.6%) underwent radical prostatectomy (n = 34 patients underwent concomitant lymphadenectomy: n = 29 only contralateral to the graft side, n = 2 bilateral but less extended on the graft side, and n = 1 complete bilateral), followed by radiation therapy (n = 28, 16.9%) and AS (n = 17, 10.2%) (see Appendix A).

### 3.3. Oncological Outcomes and Survival

At a median follow-up of 60.5 months (IQR 31–106), 22.9% of men (n = 38) died, with only n = 4 deaths due to PCa. Amongst surviving patients, the majority were without disease (62.6% of the total cohort) (see Table 3). Local and systemic progression were relatively rare, being experienced by 4.2% and 3.0%, respectively. Five patients ended active surveillance: two because of PSA rise, and three because of biopsy showing an increased Gleason score compared to the baseline assessment. Amongst patients with low-risk disease (n = 61) according to NCCN (National Comprehensive Cancer Network) criteria, one man had BCR after radiotherapy, and two patients had interrupted active surveillance. No systemic progression was detailed. Amongst men with intermediate-risk disease according to NCCN classification, three patients received active surveillance, nine patients had BCR, and two experienced systemic progression. From PCa diagnosis, 5 and 10 year OS and CSS were 83% and 42%, and 98% and 93%, respectively. Kaplan–Meier cuves for BCR/progression free survival and overall survival are detailed in Appendix A. The 5 year BCR/progression-free survival and overall survival were 0.83 (95% CI: 0.76–0.88) 5 and 0.86 (0.79–0.91), respectively.

### 3.4. Univariable Analysis

On univariable analysis (Table 4), pretransplant immunosuppression (HR 5.58, 95% CI 1.42–22, *p* = 0.01) and PSA (1.14, 95% CI 1.06–1.23, *p* < 0.01) were the only factors related to an increased likelihood of ISUP > 3 at prostate biopsy.

Need for dialysis before transplant (HR 0.35, 95% CI 0.14–0.92, *p* = 0.003), use of antiproliferative agents (HR 0.39, 95% CI 0.16–0.97, *p* = 0.04), age (HR 1.06, 95% CI 1.03–1.09, *p* < 0.001), and PCa diagnostic features including PSA (HR 1.10, 95% CI 1.05–1.15, *p* < 0.001), cT (HR 2.00, 95% CI 1.22–3.28, *p* = 0.006), cN (HR 6.52, 95% CI 2.29–18.46, *p* < 0.001), and Gleason score ≥8 (HR 7.15, 95% CI 3.07–16.65, *p* < 0.001) were significantly associated with an increased risk of progression during active surveillance or BCR after treatment.

Amongst general and transplant features negatively influencing overall survival were diabetes (HR 2.45, 95% CI 1.21–5.14, *p* = 0.001) and time from transplant to PCa diagnosis (HR 1.00, 95% CI 1.00–1.01, *p* = 0.002). Use of mTOR inhibitors versus no use (HR 4.77, 95% CI 1.39–16.35, *p* = 0.001) and use of sirolimus versus no use (HR 4.81, 95% CI 1.40–16.48, *p* = 0.001) were also associated with decreased survival. Lastly, amongst PCa-related factors, PSA (HR 1.07, 95% CI 1.01–1.14, *p* = 0.0015) and Gleason score ≥8 (HR 3.39, 95% CI 1.58–7.30, *p* = 0.002) at diagnosis were also inversely associated with survival.

## 4. Discussion

To our knowledge, this is the largest series of diagnosed with cM0 prostate cancer in kidney transplants recipients. Several findings are of interest.

First, baseline PCa features in RTR were seemingly in line with those of PCa in the general population, with mainly low- to intermediate-risk organ-confined disease being diagnosed. Specifically, almost nine out of ten patients had organ confined disease and favorable Gleason score, with median PSA also being relatively low. Of note, PSA level was likely not significantly influenced by immunosuppression drugs (e.g., sirolimus, which can lower PSA levels by as much as 50% [12]), which were used only by a minority of patients. This confirms the finding of previous institutional and population-based studies, including up to 50% of cases with a Gleason score of 6 and rarely reporting the presence of extra-prostatic spread [5,8,10]. Importantly, all patients performed routine PSA and DRE at the time of renal transplant, and the likelihood of the disease being already present at that time was low in our view. This was further supported by the almost 10 years of median time from transplant to PCa diagnosis.

Second, we provided a snapshot of the preferred treatment modalities used at seven high-volume PCa and transplant tertiary referral centers. Indeed, having included mainly urological departments, a bias toward surgical management and toward localized disease may be present; currently, as per PCa overall, no advantages of this option over the others were detailed in RTR, and the optimal management strategy remains an unanswered question that requires further research. To our knowledge, we are also the first to detail active surveillance in RTR, which was the initial management strategy in one in 10 men. Older reports suggested an increased PCa aggressiveness in RTR, thus possibly benefitting from upfront aggressive management [11]. Contrarily, we do believe that, as with the general population, surveillance can be offered to appropriately selected patients. Detailed outcomes of this and other treatment subgroups will be further described in a separate analysis.

Third, RTRs had non-negligible baseline comorbidities with more than half yielding severe systemic disturbance according to the ASA classification, despite a relatively young age. Renal function was generally low at PCa diagnosis; almost one in four men had diabetes, and one in four were previously diagnosed with other concomitant malignancies. Indeed, these aspects must be carefully considered in PCa decision making, especially when deciding whether active treatment should be performed and when envisaging the appropriate treatment type. Although comorbidities of RTR are well detailed, previous studies investigating PCa in RTRs poorly reported this aspect [5,8,11]. Another relevant aspect which should be taken into account is the high number of concomitant immunosuppression-related and non-immunosuppression-related malignancies diagnosed in these men [13]. Although the majority of data derive from radical surgery, given the high baseline in RTRs, less invasive modalities such as radiotherapy or ablation should be considered and require further investigation/assessment in this setting.

Fourth, at 5 years follow-up, PCa-specific survival was reasonable, whilst overall survival was not, with one in five men dying due to PCa-unrelated causes. In terms of PCa-related mortality, these results are similar to those of a typical mainly low- to intermediate-risk PCa cohort [14]. In terms of overall mortality, data are also in line with the reported life expectancy for RTRs [15,16]. These findings are consistent with the majority of previous studies on PCa in RTRs [5,8]. Interestingly, a previous report investigating 62 RTRs diagnosed with PCa found similar overall survival, although the mean follow-up was only 2 years; more than half of deaths were due to PCa [11]. Higher rates of locally advanced PCa and inclusion of some patients with systemic disease may only in part explain these differences.

Fifth, we investigated possible factors associated with worse or improved outcomes of PCa in RTR. Interestingly, receiving immunosuppression before kidney transplant was associated with an increased risk of Gleason score >7 but was not related to a worse natural course of PCa or to increased risk of death. Moreover, no major influence of different immunosuppression regimens on PCa was found. Despite a mildly protective effect of antiproliferative agents, including mycophenolate mofetil and azathioprine, against PCa BCR or progression after active surveillance, no effect of these two drugs alone was found, and this association remains unclear. Similarly, other types of association not previously described remain to be elucidated.

From a clinical perspective, we confirmed the hypothesis that PCa in RTR does not behave more aggressively compared to the general population, with the largest series available to date. The majority of PCa yields a low to intermediate risk, is organ-confined at diagnosis, and has a relatively slowly evolving natural course with high control and curative rates [5,8,10]. Furthermore, RTRs have a lower life expectancy compared to the general population and have a non-negligible risk of death, more likely due to causes other than PCa. Hence, indiscriminate aggressive upfront management of PCa in RTRs should be avoided, and the risk of overtreatment in this population must be even more carefully considered than usual. 

From a research perspective, we highlighted some baseline and PCa features which may influence disease and patient natural history including the immunosuppression regimen being used, which, contrarily to a previous report, does not seem to play a major role. Overall, despite some positive associations being found on univariate analysis, further work is indeed needed due to study limitations. The retrospective nature of the study may have hampered data quality, and some PCa cases may also have been missed. Furthermore, the relatively low number of patients and PCa-related events did not allow multivariate analysis, with the effect of relevant confounding factors, including treatment modality, remaining unquantified.

Hence, no major conclusions on factors possibly influencing PCa course in RTRs should be drawn from our work. Efforts should be made to evaluate the role of patient- and graft-related factors; this should be applied not only in relation to PCa natural history but also, together with PCa features, in relation to patients’ life expectancy, possibly leading to nomograms minimizing both over- and undertreatment.

As highlighted by others [5,8], future work in this field should consist of multicenter prospective data collection, promoted by national and international medical associations, which are the methods via which larger cohorts may be generated.

## 5. Conclusions

PCa diagnosed in RTRs is mainly of low to intermediate risk and organ-confined at diagnosis, showing a relatively slowly evolving natural course with good cancer control and low PCa death at intermediate follow-up. RTRs have a non-negligible risk of death, more likely due to causes other than PCa. Indiscriminate aggressive upfront management of the majority of RTRs with PCa should be avoided.

## Figures and Tables

**Table 1 cancers-15-00189-t001:** Patients and kidney transplant baseline features.

Patients and Transplant Features	n (%)/Median (IQR)	
n	166	(100.0)
Patent baseline features		
Age	67	(60–73)
Race		
Caucasian	115	(79.3)
Afro-american	8	(5.5)
Asiatic	22	(15.1)
BMI	26	(24–29)
Diabetes	33	(23.9)
ASA score		
1	1	(0.9)
2	46	(40.7)
3	62	(54.9)
4	4	(3.5)
Malignancies other than PCa	34	(24.1)
Transplant and kidney Failure features		
Renal function at PCa diagnosis		
Creatinine (mg/dL)	1.7	(1.27–2.39)
eGFR (mL/min)	45.9	(31.5–63.4)
Renal failure		
Chronic	143	(98.6)
Acute	2	(1.4)
Cause of renal failure		
Chronic glomerulonephritis	50	(34.7)
APKD	28	(19.4)
Diabetic nephropathy	12	(8.3)
Nephrosclerosis	17	(11.8)
Urate nephropaty	1	(0.7)
Vesicoureteral reflux	3	(2.1)
Chronic pyelonephritis	2	(1.4)
Congenital renal dysplasia	3	(2.1)
Others	28	(19.4)
Previous dialysis	113	(79.6)
Hemodialysis	91	(83.5)
Peritoneal	12	(11.0)
Both	6	(55.0)
Number of kidney transplants		
1	124	(84.9)
2	19	(13.0)
>2	3	(2.0)
Time first transplant to PCa (mo)	117	(48–192)
Type of first transplant		
Single cadaver	112	(79.4)
Singe living donor	25	(17.7)
Double cadaver	4	(2.8)
Other transplanted organs	4	(2.4)
Liver	2	(1.2)
Heart	1	(0.6)
Pancreas	1	(0.6)
Immunosuppression		
mTOR inhibitors	9	(5.4)
Antiproliferative agents	104	(62.6)
Calineurin inhibitors	117	(70.5)
Steroids	57	(34.4)
Time IS to T (months)	129	(49–250)

Time IS to T = time from the beginning of immunosuppression to PCa treatment; APKD = autosomal dominant polycystic kidney disease; ASA = American Society of Anesthesiologists; BMI = body mass index. Percentages are calculated for the total number of patients with detailed variables available.

**Table 2 cancers-15-00189-t002:** PCa baseline features.

PCa Baseline Features	n (%)/Median (IQR)	
Patent PCa features		
PCa familiarity *	8	(6.0)
PSA (ng/mL)	6.5	(5.02–10)
Prostate biopsy		
Biopsy Gleason score		
6	70	(42.7)
7	73	(44.5)
≥8	21	(12.8)
Biopsy cores		
Taken	12	(12–18)
Positive	4	(2–7)
mpMRI		
Preop mpMRI		
No	48	(34.8)
Yes	99	(65.2)
With contrast	81	(90.0)
Without contrast	9	(10.0)
Findings		
Negative	13	(13.1)
Positive (index lesion)	68	(68.7)
Positive (>1 lesion)	18	(18.2)
ECE	14	(14.1)
SVI	2	(2.0)
Clinical stage		
Clinical T-stage		
≤2	145	(88.4)
3	18	(11.0)
4	1	(0.6)
Clinical N-stage		
0	154	(93.9)
1	10	(6.1)
NCCN risk stratification		
Low	61	(37.2)
Intermediate	71	(43.3)
High	32	(19.5)

ECE = extracapsular extension; SVI = seminal vesicle invasion; CT and or bone scan negative for metastasis; n = 15 also had a baseline PET scan negative for extra-prostatic extension, n = 12 choline, and n = 3 PSMA. Family history of PCa defined as ≥2 first- or second-degree relatives with PCa on the same side of the pedigree. NCCN = National Comprehensive Cancer Network criteria, missing for n = 2 patients.

**Table 3 cancers-15-00189-t003:** Oncological outcomes of kidney transplant patients being diagnosed with cM0 prostate cancer.

Oncological Outcomes			
Follow-up (months)	60.5		(31–106)
Status at last follow-up—*n (%)*			
Dead	* 38 *		* (22.9) *
Non-PCa-related	33		(19.9)
PCa-related	4		(2.4)
Graft related	1		(0.6)
Alive	* 128 *		* (77.1) *
No PCa	104		(62.6)
BCR without T	10		(6.0)
BCR with ADT	9		(5.4)
BCR NA	5		(3.0)
**Disease natural history *—** *n (%)/median (IQR)*			
**BCR/progression**	27	/149	(18.1)
Time (months)	20		(3.5–46.5)
**Local progression**	7	/166	(4.2)
*Time (months)*	21		(17–94)
**Systemic progression**	5	/166	(3.0)
*Time (months)*	27.5		(12.5–45)

* Time to the event being available; BCR NA= biochemical recurrence with no info on whether the patient underwent treatment or not; T = treatment; ADT = androgen deprivation therapy. Of those with BCR/progression, n = 17 were in a post-radical prostatectomy setting, n = 6 had post- radiotherapy, and n = 4 received post-active surveillance.

**Table 4 cancers-15-00189-t004:** Univariate analysis.

	Gleason Score ≥ 4			BCR/Progression			Death		
	OR	95% CI	*p*	HR	95% CI	*p*	HR	95% CI	*p*
Diabetes	0.81	0.25–2.63	0.7	0.64	0.18–2.25	0.5	2.45	1.21–5.14	0.001
Renal failure	-	-	-	-	-	-	0.18	0.02–1.36	0.1
Chronic glomerulonephritis **	0.66	0.25–1.71	0.4	0.93	0.39–2.20	0.9	0.53	0.26–1.08	0.9
Dyalisis needed	0.75	0.25–2.27	0.6	0.35	0.14–0.92	0.003	0.70	0.33–1.50	0.4
Hemodyalisis ^§^	0.35	0.06–2.17	0.3	0.49	0.009–2.60	0.4	0.91	0.41–2.02	0.8
Number of transplants >1	1.46	0.44–4.86	0.6	1.75	0.57–5.35	0.3	2.01	0.87–4.61	0.1
Time from T to D (mo)	1.00	0.99–1.00	0.2	1.00	0.99–1.00	0.4	1.00	1.00–1.01	0.002
Pretransplant IS (mo)	5.58	1.42–22	0.01	1.02	0.22–4.67	0.9	-	-	>0.99
Immunosuppression									
mTOR inhibitors	2.17	0.41–11.57	0.4	2.78	0.62–12.38	0.2	4.77	1.39–16.35	0.001
Antiproliferative agents	0.71	0.26–1.93	0.5	0.39	0.16–0.97	0.04	1.01	0.49–2.09	0.9
Calcineurine inhibitors	0.92	0.28–3.01	0.9	0.81	0.26–2.46	0.7	1.22	0.43–3.50	0.7
Steroids	0.81	0.30–2.18	0.7	0.70	0.26–1.89	0.5	1.02	0.53–1.98	0.9
Tacrolimus	1.04	0.37–2.91	0.9	0.64	0.21–1.94	0.4	1.04	0.50–2.15	0.9
MMF	0.75	0.28–1.98	0.6	0.42	0.17–1.04	0.06	1.20	0.58–2.49	0.6
Ciclosporine	1.68	0.59–4.81	0.3	1.96	0.74–5.17	0.2	1.66	0.82–3.37	0.2
Everolimus	-	-	>0.9	-	-	>0.9	-	-	>0.9
Azatioprine	0.87	0.10–7.49	0.9	0.93	0.12–7.06	0.9	-	-	>0.9
Sirolimus	1.03	0.12–9.00	0.9	2.94	0.66–13.10	0.2	4.81	1.40–16.48	0.001
Others	3.18	0.28–36.9	0.4	4.19	0.55–31.92	0.2	-	-	>0.9
Age at treatment	0.98	0.94–1.02	0.3	1.06	1.03–1.09	<0.001	0.96	0.94–0.99	0.0001
PSA at diagnosis	1.14	1.06–1.23	<0.001	1.10	1.05–1.15	<0.001	1.07	1.01–1.14	0.0015
cT	-	-	-	2.0	1.22–3.28	0.006	1.55	0.98–2.44	0.06
cN	-	-	-	6.52	2.29–18.46	<0.001	2.09	0.72–6.02	0.2
Gleason Score ≥4	-	-	-	7.15	3.07–16.65	<0.001	3.39	1.58–7.30	0.002

Univariate analysis on factors possibly influencing PCa histological aggressiveness at diagnosis, BCR/progression, and/or overall survival. Due to the low number of recorded events, no univariate analyses were performed for local and/or systemic progression and PCa-related deaths. Similarly, due to the low number of events, no multivariate analyses were performed. Pretransplant IS (immunosuppression): “having immunosuppression before transplantation, generally due to an underlying cause of renal insufficiency (e.g., glomerulonephritis)”. Immunosuppression: drug analyses on univariate analysis were performed considering those using the drug or drug category versus those not using it. ** Chronic glomerulonephritis versus other causes of renal failure; ^§^ hemodialysis versus others; T to D, time from transplant to diagnosis.

## Data Availability

Data are available upon a specific request to the corresponding author.

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
