# Peer review of "Prostate Cancer in Renal Transplant Recipients: Results from a Large Contemporary Cohort"

_cancers, 2022, doi:10.3390/cancers15010189_

Round 1
Reviewer 1 Report
Marra and colleagues present a sound manuscript on the behavior and outcomes of patients who suffer from prostate cancer after renal transplantation. The authors have collected a relatively large cohort, considering the rare incidence. Therefore, this manuscript provides interesting insights for this rarely investigated patient cohort. As the authors state in the manuscript, further analysis of the collected cohort are planned in future.
I have only minor comments:
1) Please provide NCCN risk group at baseline patient characteristics
2) Please provide a Kaplan Meier curve for BRFS and OS for the cohort. I think this would help to visualize your results.
3) As you point out you do not recommend aggressive therapies for the majoriry of patients. Nevertheless, in your cohort most of the patients were treated with surgery. Considering the high ASA scores of these patients and comorbidities, non-invasive treatment options such as EBRT might be of particular interest for these patients. I would suggest elaborating this aspect more in the discussion.
Reviewer 2 Report
This is a multicentre retrospective study of men who had kidney transplant and subsequently diagnosed with prostate cancer. 166 men from 9 European centres diagnosed with prostate cancer between 2001 and 2019 were included in the study. Overall, laudable effort collating probably the largest series in this cohort, and will be good addition to the literature. A major finding that stands out (and also highlighted by the authors in discussion) is that the 5- and 10-year OS were 83% and 42% respectively (but the prostate cancer specific mortality is low), and that it’s important to make sure we don’t over- or under-treatment prostate cancer in patients with kidney transplant.
However, there are several comments/ suggestions below (especially strongly recommend formal statistical input).
- Line 63, why were the men with metastatic prostate cancer excluded?
- 2.3 covariables definitions – I assume that the 2 patients who had WW were not included in BCR/ progression outcomes, if so, please specify. Also the 6 patients on ADT, are they long-term ADT, and how do you define BCR/ progression for them, or are they included in BCR/ progression outcomes?
- Statistics – BCR should be analysed as time to event, rather than binary outcomes (yes/no) with logistic regression.
- Table 1 – time first transplant to prostate cancer diagnosis – repeated again in Table 2 – please remove from either one of the tables.
- Table 2 – ‘PCa familiarity’ – what does this mean? Family history of prostate cancer? please reword for clarity
- Table 2 – please include a row on the number and proportion who had NCCN low/ int/ high risk – it’s mentioned somewhere in the text (line 140) that there were 61 low risk , but no information about the rest.
- Table 2 – cM stage: this row is meaningless as the authors have excluded the patients with metastatic disease from the study cohort.
- Supplementary material – for the primary treatment modalities, please stratify by NCCN risk (i.e., separate columns for low/ int/ high risk), so readers can understand how the different risk group are managed for patients with kidney transplant e.g. are all the RT given to high risk? Also the RT group, whether RT was given with concurrent ADT. And the 6 who had ADT are primary ADT alone, and are these patients with cN1 disease?
- Table 3 – disease natural history row – please include denominator in your table e.g. BCR 27/ XXX (18.1%). The proportion is incorrect if I take all your active treatment patients as denominator. Same apply to local progression and systemic progression.
- Table 4 – covariables (Column 1): For the baseline features e.g., cause of renal failure, it appears that this is analysed as continuous variable with a single OR 0.66 (95%0.25-1.71), but in Table 1 there are a whole range of causes of renal failure. This is inappropriate analyses for this variable, as this is not a continuous or ordinal variable (and which is the reference group e.g. chronic glomerulonephritis?). This applies to the variable on the type of Dialysis. For the rows on the various immuno-suppressants, what are this OR/ HR compared against (i.e. what is the reference group)? Please recheck/ clarify the statistical analyses method. Time from transplant to diagnosis (T to D) – please include explanation for the abbreviation as footnote. Also, when you look at HR for death for T to D (HR=1.00, 95%CI=1.00-1.01; P=0.002), this is not meaningful for interpretation when you have lower confidence interval that essentially the same as the HR. Recommend analysed as e.g. every 6 or 12 months categories especially you have median T to D of 117 months. cT/ cN – very small number with cN1 (6%); may be worth looking into NCCN low/ int/high risk (instead of cT/ cN). ISUP – recommend analyses as ordinal categories (rather than continuous)
- Table 4 – outcomes on ISUP>=4 (column 2). In methods, it’s stated that the outcome is ISUP>=3, but in table it is ISUP>=4. Please clarify which is correct. Also, Outcomes for logistics regression should be OR not HR (on the first row of table). Also important to know what’s the proportion who had ISUP 1/2/3/4/5. Table 1 reported Gleason 6/7/>=8 whereas Table 4 reported ISUP grade group. Recommend consistency in Gleason or ISUP grade group reporting (pick one and be consistent across Tables).
- Table 4 – outcomes on BCR/ progression (column 3) – as per comments on analyses on BCR above; this should be analysed using time-to-event rather yes/no binary outcomes
- Discussion – paragraph 2 – the men captured in the current cohort is probably just the type of patients seen/ treated collectively by the authors, and the authors have intentionally excluded men with metastatic prostate cancer (though small in number), so, can’t really claim that prostate cancer features in kidney transplant patients are similar to general population (it’s just the patients captured in the authors’ institutions that may be similar to general population). This is highlighted in the discussion paragraph 3 whereby the authors are largely capturing patients in urology services, and hence likely capturing more patients who were fit for/ referred for surgery.
Reviewer 3 Report
Dear Authors,
thanks for sharing your data on this topic.
The text, while reading, flows easily enough. The tables can be improved from an aesthetic and functional point of view because in this way they are not, however, so clear and legible. I leave you some specific changes:
Page 2 line 52 (and later in the paper): i would use “patients” instead of “men”
Page 2 line 56: “DRE” has no written definition of digital rectal exploration
Page 2 line 62 : “Metastatic 62 patients were excluded (n=5)”—> could you please explain in details why you didn’t include those patients in the study?
Page 2 line 62 : BCR has no definition (it has in line 74 but must be added here)
Page 3 line 102: “n=34 (24.1%) had a history of malignancies other than PCa”. It seems to me a very high number. It should be explained, at least in the discussion
Page 7 line 156: “On univariable analysis (Table 4) pre-transplant immunosuppression” I think you meant “pre-PCa immunosuppression”. You repeated in the table as well.
Page 7 line 159: “Use of mTOR inhibitors (HR 4.77, 95% CI 1.39-169 16.35, p=0.001) and of sirolimus (HR 4.81, 95% CI 1.40-16.48, p=0.001) were also associated 170 with decreased survival”.
This sentence has the error of distinguishing m-TORi and sirolimus, when the latter is part of the m-TORi themselves. Perhaps the authors intended to distinguish Everolimus and Sirolimus. Also, talking about worse survival and not tumor recurrence doesn't make much sense in this part of the article.
